# Ovarian Cancer: Biomarkers and Targeted Therapy

**DOI:** 10.3390/biomedicines9060693

**Published:** 2021-06-18

**Authors:** Mihaela Raluca Radu, Alina Prădatu, Florentina Duică, Romeo Micu, Sanda Maria Creţoiu, Nicolae Suciu, Dragoş Creţoiu, Valentin Nicolae Varlas, Viorica Elena Rădoi

**Affiliations:** 1Fetal Medicine Excellence Research Center, Alessandrescu-Rusescu National Institute for Mother and Child Health, 020395 Bucharest, Romania; ralucamihaelaradu@gmail.com (M.R.R.); neagualina0206@gmail.com (A.P.); florentina.duica80@gmail.com (F.D.); nsuciu54@yahoo.com (N.S.); 2Department of Mother and Child, Iuliu Hatieganu University of Medicine and Pharmacy, 400012 Cluj-Napoca, Romania; romeomicu@hotmail.com; 3Department of Cell and Molecular Biology and Histology, Carol Davila University of Medicine and Pharmacy, 050474 Bucharest, Romania; sanda@cretoiu.ro; 4Division of Obstetrics, Gynecology and Neonatology, Carol Davila University of Medicine and Pharmacy, 050474 Bucharest, Romania; 5Department of Obstetrics and Gynecology, Alessandrescu-Rusescu National Institute for Mother and Child Health, Polizu Clinical Hospital, 020395 Bucharest, Romania; viorica.radoi@yahoo.com; 6Department of Obstetrics and Gynecology, Filantropia Clinical Hospital, 01171 Bucharest, Romania; 7Faculty of Dental Medicine, Carol Davila University of Medicine and Pharmacy, 030167 Bucharest, Romania; 8Department of Medical Genetics, Carol Davila University of Medicine and Pharmacy, 050474 Bucharest, Romania

**Keywords:** ovarian cancer, biomarkers, ncRNAs, targeted therapy, PARP inhibitors, NTRK inhibitors

## Abstract

Ovarian cancer is one of the most common causes of death in women as survival is highly dependent on the stage of the disease. Ovarian cancer is typically diagnosed in the late stage due to the fact that in the early phases is mostly asymptomatic. Genomic instability is one of the hallmarks of ovarian cancer. While ovarian cancer is stratified into different clinical subtypes, there still exists extensive genetic and progressive diversity within each subtype. Early detection of the disorder is one of the most important steps that facilitate a favorable prognosis and a good response to medical therapy for the patients. In targeted therapies, individual patients are treated by agents targeting the changes in tumor cells that help them grow, divide and spread. Currently, in gynecological malignancies, potential therapeutic targets include tumor-intrinsic signaling pathways, angiogenesis, homologous-recombination deficiency, hormone receptors, and immunologic factors. Ovarian cancer is usually diagnosed in the final stages, partially due to the absence of an effective screening strategy, although, over the times, numerous biomarkers have been studied and used to assess the status, progression, and efficacy of the drug therapy in this type of disorder.

## 1. Introduction

Even though medicine has made amazing progress in the last few years, ovarian cancer still remains a challenge around the world. Ovarian cancer is one of the most frequent causes of death in women as survival is highly dependent on the stage of the disease. Ovarian cancer is typically diagnosed in the late stage due to the fact that in the early phases is mostly asymptomatic [1]. According to the literature, there are different subtypes of ovarian cancer. Based on the morphology of tumor cells, the ovarian cancer was divided by histological subtype as serous, endometrioid (EC—endometrioid carcinoma), mucinous (MC—mucinous carcinoma), with clear cells and squamous cells (CCC for clear cell carcinoma and SCC for squamous cell carcinoma) [2,3,4,5]. The genetics of ovarian cancer are a complex, ever-evolving concept that presents hurdles in classification, diagnosis, and treatment in the clinic. Instead of common driver mutations, genomic instability is one of the main features of ovarian cancer. While ovarian cancer is stratified into different clinical subtypes, there still exists extensive genetic and progressive diversity within each subtype.

Early detection of the disorder is one of the most important steps that facilitate a good prognosis and a good response to medical therapy for the patients. The centerpiece that contributes to an early diagnosis may be represented by non-invasive prognostic biomarkers, like non-coding RNAs (microRNAs, long noncoding RNAs, circular RNAs, and transfer RNA-derived small non-coding RNAs) [6]. 

The present review comprises the latest findings in ovarian cancer research with the purpose of a better comprehension of this complex disease.

## 2. Classification and Histopathology

According to numerous published studies, the term ovarian cancer is generally attributed to a number of diseases that are distinct in terms of etiology and molecular characteristics, but which simply share an anatomical appearance [5,7,8,9,10,11]. 

Along with advances in pathological and genomic diagnostic technology, major progress has been made in understanding the cellular and molecular biology of human cancers. Recent findings have indicated that several types of ovarian cancer classified into different histotypes are actually derived from non-ovarian tissues and share few molecular similarities. Ovarian cancer is a multifactorial disease that can be subdivided into at least five different histological subtypes, that have different risk factors, and manifests various clinical features, in which the cells of origin and molecular changes are multiple, and where different treatments are addressed [10,11]. The classification of ovarian cancer is generally made according to the stage at the time of tumor discovery, early or advanced stage, classified as low grade and high grade of malignancy (HGSC for high-grade serous carcinoma and LGSC for low-grade serous carcinoma), but especially depending on the histological subtype [3,4,5,12].

According to pathologists perspective, ovarian cancers classification is based on the morphology of tumor cells and was divided by histological subtype as serous, endometrioid (EC—endometrioid carcinoma), mucinous (MC—mucinous carcinoma), with clear cells, and squamous cells (CCC for clear cell carcinoma and SCC for squamous cell carcinoma) [2,3,4,5]. The understanding of ovarian cancers has evolved as a result of recent molecular research reported in large clinical trials, that revealed the importance of characteristic genetic defects for each major histological type. Although multiple different grading systems for the classification of ovarian cancer have been used, one clear image regarding the framing of a tumor in a particular subtype is difficult to establish, due to many characteristics that need to be considered [3,13].

The WHO guidelines published in 1973, was the first attempt to make a systematic classification of the many ovarian cancer subtypes based on architecture (microscopic characteristics of the tumors) and cytologic characteristics (the nature of morphologically identifiable cell types and patterns). The latest version of WHO Classification of Ovarian Cancer (published in 2014 by Robert Kurman and co-authors) takes into account new disclosed characteristics, regarding the origin of the OVC (ovarian carcinoma) tumor cells, pathophysiological mechanism (mechanisms of development and progression in ovarian cancers), pathological features, treatment response and prognosis of different ovarian cancer subtypes (see Table 1) [14]. 

FIGO system implemented in 1988 by the International Federation of Gynecology and Obstetrics and revised recently, and the AJCC (American Joint Committee on Cancer) TNM (tumor, lymph nodes, metastasis) staging system, classify ovarian cancers using 3 factors:—the size of the tumor, the spread to nearby lymph nodes, and the metastasis (spread to distant sites). It was reported in four stages, ranging from stage I through stage IV, depending on methods used for diagnosis. It could be used the surgical stage (also called pathologic stage) examining tissues obtained after chirurgical intervention, or clinical stage based on the results of multiple investigations such a physical exam, biological tests, biopsy, and imaging tests [3]. 

FIGO -the surgical staging system, was reviewed and updated in parallel with The World Health Organization systems in 2014, and both are applied to all histotypes of ovarian cancers. The changes made are meant to provide a better understanding both of the diagnosis and on the approach of the care provided and the therapeutic possibilities [31].

Another largely used grading system has been that proposed by the Gynecologic Oncology Group (GOG) where the grading methods are divided by histological type of the tumor [2,32]. 

### 2.1. Histological Classifications

The histological aspect of ovarian carcinomas is revealed by the study of the arrangement of tissues at the microscopic level and implies an identification of tissue abnormality that is important to establish the diagnostic, for clinical management of therapy and prognostic. 

As, at the beginning of therapies for OVCs, histopathological analysis is considered to be the gold-standard method for diagnosis due to its accessibility and cost relief efficiency. 

Based on this method, the most common histological types of low-grade and high-grade tumors of epithelial and non-epithelial ovarian cancers are subdivided into two main groups: Type I carcinomas and Type II carcinomas, with appellations deriving from their morphology and tissue architecture characterized by imagining techniques (microscopy). Even that the majority of diagnosed of ovarian carcinoma are included in one of the four major histotypes based on histological appearance, sites of origin, and modes of carcinogenesis or molecular-genetic features, some rarer types have been reported, such as malignant transitional cell (Brenner) carcinoma, mixed type carcinoma and undifferentiated carcinoma. Low-grade serous carcinoma (LGSC), low-grade endometrioid ovarian carcinomas (ENOC), clear cell carcinomas (CCC), mucinous carcinomas (MC), and malignant Brenner tumors are included in the first group, Type I carcinomas, and high-grade serous carcinomas (HGSC), high-grade ENOCs, undifferentiated carcinomas, and carcinosarcomas, was included in Type II carcinomas [12]. In one recent study published by Santandera et al., it was suggested the possibility to classify ovarian cancers, by a molecular-based classification where recent evidence on the molecular analysis, traditional targeted DNA sequencing, and immunohistochemistry (IHC) results are evaluated.

In accordance with The Cancer Genome Atlas Network (TCGA) published studies, it was proposed at least five main entities where molecular, histological, clinical, and pathological features are considered: HGSCs, LGSCs, ENOCs, MCs, and CCCs [12,33,34]. The importance of a molecular-based classification is sustained by the need for easily clarified diagnosis and development of target tailored therapies in particular cases of OVCs (development of personalized medicine).

### 2.2. Phenotypic Classifications

According to the female gonadal structure, ovary carcinomas can be framed into one of three major categories, in the function of the anatomic structure from which the tumoral lesion presumably originates. These principal categories of tumors are surface epithelial-stromal tumors, sex cord-stromal tumors, and germ cell tumors, and each of the tumoral classes is subdivided into a number of subtypes [35]. A recent publication on morphologic, immunohistochemical, and molecular genetic studies has revealed new approaches in evaluating and managing ovarian carcinogenesis [36,37].

Elucidation of the sites of origin and progression of ovarian cancer is difficult to achieve due to the fact that these type of neoplasms are constituted of a variety of histologic subtypes, that has distinct cells of origin, biomolecular constitution, clinicopathological features, and treatments. Regardless of the site of origin, ovarian cancers frequently involve malignant changes at the peritoneal cavity, in the para-aortic and pelvic lymph nodes, but also in some distant organs such as breast, liver, lungs, or other [36,38]. 

## 3. Genetics of Ovarian Cancer

To date, the cellular origin and pathogenesis of OVCs are not well understood, but it was established that pathogenic germline mutations in BRCA1 or BRCA2 genes are the main risk factors in the development of ovarian cancer. Other genetic mutations which target specific cell signaling pathways are involved in ovarian cancers (see Table 2).

### 3.1. BRCA1 and BRCA2 Genes

Germline mutations of the BRCA1 and BRCA2 genes lead to a high lifetime risk of ovarian cancer. They represent the predominant and most well characterized genetic risk factors so far identified for the disease. BRCA1 and BRCA2 are involved in almost half of all families containing two or more ovarian cancer cases.

The BRCA1 gene is located on chromosome 11q21 and comprises 22 coding exons spanning 80 kb of genomic DNA and has a 7.8 kb transcript coding for an 1863 amino acid protein. The BRCA2 gene is located on 13q12-13 and comprises 26 coding exons, spanning 70 kb of genomic DNA, has an 11.4 kb transcript, and codes for a 3418 amino acid protein. Both proteins function in the double-strand DNA break repair system.

About 1.2% of women in the general population will develop ovarian cancer sometime during their lives (1). By contrast, 39–44% of women who inherit a pathogenic BRCA1 variant and 11–17% of women who inherit the pathogenic BRCA2 variant will develop ovarian cancer by 70–80 years of age [76]. 

A patient’s prognosis for BRCA1/2-related cancer depends on the stage at which the cancer is diagnosed and on the type of mutation; however, studies of survival have revealed conflicting information for individuals with germline BRCA1 or BRCA2 pathogenic variants when compared to controls. Retrospective studies suggest that heterozygosity for a BRCA hereditary pathogenic variant in ovarian cancer patients is associated with a significantly more favorable prognosis and is predictive of sensitivity to combination therapies containing platinum derivatives [77,78] whereas others have shown the opposite [79,80].

Evidence exists that ovarian cancer patients carrying germline BRCA mutations have a better prognosis and overall survival when compared to sporadic cases [81,82].

As the BRCA1 and BRCA2 genes codify for proteins that are involved in DNA repair, tumors with alterations in either gene are particularly sensitive to specific anticancer agents that act by damaging DNA [83].

### 3.2. Other Genes

Several tumor suppressor genes and oncogenes have been associated with ovarian cancers, including the p53 tumor suppressor gene, the mismatch repair (MMR) genes, and few other genes involved, along with BRCa1 and BRCA2 in the double-strand breaks repair system, such as CHEK2, RAD51, BRIP1, and PALB2.

A significantly increased risk of ovarian cancer is also a feature of certain rare genetic syndromes, including Lynch syndrome and Li Fraumeni. Lynch syndrome is most often associated with mutations in the MLH1 or MSH2 gene and Li Fraumeni is caused by a germline mutation in the p53 gene.

#### 3.2.1. MMR Genes

The mismatch repair (MMR) system is a mechanism that corrects mutations arising during DNA replication or damage, and it has a crucial role in maintaining genome stability [84,85]. MMR system is a comprehensive pathway involving key components at each phase. Seven MMR genes, mutL homolog 1 (MLH1), mutL homolog 3 (MLH3), mutS homolog 2 (MSH2), mutS homolog 3 (MSH3), mutS homolog 6 (MSH6), postmeiotic segregation increased 1 (PMS1), postmeiotic segregation increased 1 (PMS2) are involved in human MMR system. It is now very well-known that the inactivation of MMR in human cells is associated with genome-wide instability, including microsatellite or DNA damage, predisposition to certain types of cancer [86,87]. 

In ovarian cancer, MMR deficiency is the most common cause of hereditary ovarian cancer after BRCA1 and BRCA2 mutations [88]. A high proportion of ovarian cancers from women who have germline mutations in mismatch repair genes demonstrate microsatellite instability (MSI), but the clinical utility of pre-screening ovarian cancer tumors for MSI to identify potential patients for germline screening for MMR mutations is still uncertain.

#### 3.2.2. CHEK2 Gene

CHEK2 is a tumor suppressor gene localized to human chromosome 22 (22q12.1), where it spans 54 kb (chr22: 28,687,743–28,742,422; reverse strand; GRCh38). The most expressed transcription variant 1 (NM_007194/ENST00000404276.6) codes for an mRNA consisting of 15 exons with the translation start localized in exon 2. The relevance of alternative splicing variants remains unclear, but their proportion increases in tumor tissues. CHEK2 gene that encodes a protein kinase activated in response to DNA damage and has also been shown to interact with BRCA1, promoting cellular survival after DNA damage [89,90]. 

The role of CHEK2 mutations in ovarian cancer cancerogenesis is well known. Particularly, the missense variant of CHEK2 I157T was significantly associated with ovarian cystadenomas, borderline ovarian tumors, and low-grade invasive cancers but not high-grade ovarian cancer [89].

#### 3.2.3. Somatic Mutations in Ovarian Cancer

In the era of precision medicine, the identification of several predictive biomarkers and the development of innovative therapies have dramatically increased the request for tests to identify specific targets on cytological or histological samples, revolutionizing the management of the tumoral tissues. 

Among 4 of the more frequent cancers in women (breast, ovarian, endometrial, and cervical cancers), PTEN represents one of the most frequently mutated genes (13%) [91].

PTEN mutations can co-exist and lead to PI3K/Akt/mTOR pathway aberrantly activation; the combination of PTEN mutations with KRAS ones in the ovary has been shown to induce invasive and metastatic endometrioid ovarian cancer. PTEN is a tumor suppressor gene on chromosome 10 (cytogenetic location 10q23.3) and is variably mutated and/or deleted in several variated human cancers. Among several series of ovarian cancers, the frequency of loss of heterozygosity (LOH) of markers flanking and within PTEN, is 30 to 50%, and the somatic PTEN mutation frequency is <10% [92,93].

Tropomyosin receptor kinase (TRK) is a receptor in the tyrosine kinase family that is activated by neurotrophins, a family of nerve growth factors. Three members of the TRK family have been described: TRKA, TRKB, and TRKC, encoded by neurotropic tropomyosin receptor kinase 1 (NTRK1), NTRK2, and NTRK3, respectively [94]. The NTRK1, 2, and 3 genes encode a family of tyrosine kinase receptors with an active role in neural development. All rearrangements cause constitutive activation of these proteins. NTRK rearrangements have been reported in a series of solid and hematological tumors, with variable frequencies. These recent discoveries raise diagnostic and therapeutic challenges [95]. 

The Food and Drug Administration (FDA) has recently approved a selective neurotrophic tyrosine receptor kinase (NTRK) inhibitor, larotrectinib. Contemporarily, the development of multi-kinase inhibitors with activity in tumors carrying TRK fusions is ongoing. Chromosomal translocations involving the NTRK1, NTRK2, and NTRK3 genes result in constitutive activation and aberrant expression of TRK kinases in numerous cancer types [96].

## 4. Role of Microenvironmental Factors in Ovarian Cancer

Ovarian Cancer is a heterogeneous medical condition and is influenced by genetic and epigenetic factors [97]. 

A major reason for the lack of success in effectively curing ovarian cancer can be due to the complex interconnected signaling pathways in conjunction with the distinctive peritoneal tumor microenvironment. Some immune cells, including tumor-associated macrophages, T cells, natural killer cells in conjunction with fibroblasts, and a wide spectrum of chemokines and cytokines all interact with each other to promote the tumor cells’ growth and metastasis [98]. There is an increasing knowledge of the role that the tumor microenvironment—consisting of tumor cells, surrounding stromal cells, and stromal elements—has in promoting and sustaining ovarian cancer chemoresistance, recurrence, and metastasis [99]. Also is very well known that cancer cells can induce a reactive fibroblast phenotype, termed cancer-associated fibroblasts [100]. The functions of fibroblasts include production and deposition of types I, III, and V collagen and fibronectin, which are the most important components of the fibrillar extracellular matrix as well as the synthesis of basement membrane proteins laminin and type IV collagen [101].

## 5. Biomarkers in the Management and Prognosis of Ovarian Cancer

### 5.1. Traditional Biomarkers: CA125 and HE4

#### 5.1.1. Cancer Antigen 125 or Carbohydrate Antigen 125 (CA125)

CA125 has been utilized as a tumor marker for more than 30 years for the diagnosis of ovarian cancer. It was also used to monitor the reaction to treatment and to identify recurrence [102].

To date, medical scientists observed elevated CA125 levels in normal physiological conditions such as pregnancy, menstruation, and, also, in numerous pathological situations as ovarian lesions, endometriosis, benign/malign tumors, or pelvic inflammatory disease. In the early malign scenarios, it was proven that this marker has a low specificity and sensitivity, with 50 percent of patients with stage I tumors remaining undetected. At this time, the CA125 marker is not anymore a recommended technique of screening and diagnostic for ovarian cancer [1,103].

#### 5.1.2. Human Epididymis Protein 4 (HE4)

HE4 is a biomarker that is currently studied for diagnosing ovarian cancer. Many types of research demonstrated that HE4 is effective in the early detection and differential diagnosis of ovarian masses, although modified HE4 concentrations can be also perceptible in postmenopausal women [104,105,106,107]. 

However, HE4 combined with CA125 seems to have a more rigorous prognostic for malignancy than either alone [108].

### 5.2. Ovarian Cancer—Associated ncRNAs—Promising Non-Invasive Biomarkers

Non-coding RNAs (ncRNAs) are a special class of RNA molecules that are not translated into functional proteins. Abnormal expression levels of ncRNAs were associated with many diseases, including different types of cancer. Among ncRNAs classes that were correlated with tumor initiation and progression are microRNAs, long noncoding RNAs, circular RNAs, and transfer RNA-derived small non-coding RNAs (tsncRNA) [109].

#### 5.2.1. microRNAs (miRNAs)

Even though medicine has made amazing progress in the last few years, ovarian cancer remains a challenge around the world. Early detection of this type of cancer is one of the most important stages that facilitates a favorable prognosis and a good response to medical therapy for the affected women. The main core that contributes to an early diagnosis may be represented by non-invasive prognostic biomarkers, such as circulating microRNAs (miRNAs) [6]. The potential of miRNA as clinical biomarkers was indicated by their specificity in post-transcriptional gene regulation and their stability over time after isolation from plasma [110].

Circulating or cell-free miRNAs are containing about 19–25 nucleotides and they are involved in the modulation of some fundamental cellular processes, such as proliferation, division, differentiation, migration, and cell death [6,111,112,113]. Recent studies revealed, also, a strong correlation between aberrant expression levels of miRNAs and carcinogenesis [114,115,116]. A research-based on RT-PCR, Western blot, and bioinformatics analysis detected for the first time a significant upregulation of miRNA-552 in ovarian malignant tumors [113]. Their findings suggested that miRNA-552 associated with the PTEN gene (Phosphatase and tensin homolog) can be used for anticipation of the patient prognosis and tumor recurrence [113,117]. 

A complex review from 2020 included an impressive number of miRNAs whose expression was altered in epithelial ovarian cancer metastasis [118]. From reported miRNAs, miR-216a also modulates the expression of PTEN, being increased in ovarian cancer tissues [118,119]. An elevated level of miRNA-552 and miRNA-216a is associated with poor survival of patients [113,118,119]. For other possible predictive biomarkers, such as miR-135a [120], miR-375 [121], miR-139 [122] and miR-584 [123], downregulated levels were reported.

Yokoi and his colleagues proposed a prediction method based on a combination of 8 miRNAs that can be helpful in clinical practice. The eight potential non-invasive biomarkers were identified using miRNA global sequencing and then, the results were validated by qRT-PCR: miR-142-3p, miR-26a-5p, let-7d-5p, miR-374a-5p, miR-766-3p, miR-200a-3p, miR-328-3p, and miR-130b-3p were deregulated in ovarian cancer patients compared with healthy control lot. Early detection of ovarian cancer may improve diagnostic performance and prognosis [124].

Another research article strengthens the hypothesis that miRNAs are valuable biomarkers in ovarian malignancy detection. Some members from the miRNA-200 family (like miR-200a, miR-200b, and miR-200c) were significantly increased in women with serous epithelial ovarian cancer [125]. More recently, Elias and colleagues corroborated miRNA sequencing data with bioinformatics algorithms, thereby developing a neural network prediction model for ovarian cancer. This neural network can recognize even the tiniest tumors and has fewer false-positive results than other methods [126,127].

#### 5.2.2. Long Noncoding RNAs (lncRNAs)

Salmena et al. suggested a competitive endogenous RNA (ceRNA) hypothesis, that considers the interaction between diverse classes of RNA [128]. Novel experimental data indicates the importance and applicability of the ceRNA regulatory network in ovarian cancer [129,130]. Based on the ceRNA model, lncRNA is involved in the regulation of the gene expression by direct binding throughout the lncRNA/miRNA/mRNA axis. In this model, it is assumed that miRNAs directly binds to both lncRNA and mRNA [130,131]. However, Tian et al. noticed a direct interaction of lncRNA WDR7-7 with mRNA of the GPR30 (G-protein coupled estrogen receptor 30) gene [132]. The LncRNAs’ contribution to the regulatory networks is leading to a better understanding of the mechanisms of oncogenesis, which is essential for a proper approach to the diagnosis and prognosis of cancers [133]. 

Other members of the RNA family with no-protein encoding ability are represented by lncRNAs. They are containing more than 200 nucleotides and play a very important role in different biological functions, and also, in malignant phenotypical changes and metastasis [134,135]. According to Gong and Zou, lncRNA FAM83H-AS1 was highly expressed in ovarian cancer tissues when compared with control tissues. Their results indicate that lncRNA FAM83H-AS1 could be a veritable molecular biological index and be used to evaluate ovarian cancer progression [136]. 

The lncRNA H19 is another potential biomarker and it was identified in the majority of cases of ovarian cancer ascites fluid. The expression level of lncRNA H19 was increased in tumor samples, which may suggest the possible association of this lncRNA with carcinogenesis mechanisms [137,138]. 

Various assays demonstrated that LncRNA HOTAIR was abnormally expressed in ovarian carcinomas; an upregulated expression is directly linked with the locomotion capacity of cancer cells. When HOTAIR was knocked down, the proliferation, invasion, and migration ability of cancer cells were inhibited [139]. Chang et al. highlighted in their study the importance of the HOTAIR-miR-206-CCND1/CCND2 network in the molecular puzzle of ovarian tumorigenesis. The two cancer promoters, respectively CCND1 and CCND2, were overexpressed in the ovarian cancer samples, their expression is directly correlated with the upregulated level of HOTAIR. The elevation of HOTAIR inhibited the expression of miR-206 and elevated the expression of CCND1 and CCND2, indicating that the regulatory network previously mentioned is a great biomarker for progression and prognosis of ovarian cancer, and, why not, a new therapeutic guide [140]. The effect of lncRNA HOTAIR in ovarian carcinomas was well investigated, another potential regulatory network being described by Zhang’s team. HOTAIR-miR-373-Rab22a can represent another therapeutic target. Rab22a expression is manipulated by LncRNA HOTAIR via sponging miR-373, which consolidates the importance of this lncRNA in monitoring ovarian carcinogenesis [139]. 

The last lncRNA that we will discuss in this review is MALAT-1. According to Pei et al., MALAT-1 interacts with miR-22 and these players were atypically expressed in epithelial ovarian tumors. MALAT-1 facilitated epithelial ovarian cancer development through sponging miR-22 [141]. Another evidence of MALAT-1’s contribution to ovarian cancer progression is provided by Zhou and his team. They observed that MALAT-1 is significantly upregulated in ovarian tumors, the expression being correlated with FIGO stages [142]. MALAT-1 can be used as a molecular marker in various types of malignancies, including the most fatal gynecological cancers [143].

#### 5.2.3. Circular RNAs (circRNAs)

CircRNAs are a class of ncRNA molecules that create a covalently closed-loop structure, but without terminal structures like 5′ cap and 3′ poly-A tail [144]. Their circular form makes them more resistant against endonuclease and therefore more stable compared with linear RNAs [145]. Accumulating evidence in recent years has demonstrated the circRNA clinical significance in a vast variety of human disorders, including ovarian cancer [146]. 

As is demonstrated in recent studies, the expression level of circ-ITCH is decreased in ovarian cancer tissues, this abnormal aspect being closely associated with various clinicopathological characteristics like tumor size, FIGO, TNM stage, and also, with survival rate. Cancer research experiments supposed that circ-ITCH could act as a tumor suppressor, supporting cellular apoptosis and inhibiting tumor cell unregulated proliferation, migration, and invasion [147].

CircPLEKHM3 originated from the PLEKHM3 parental gene is one of the most notably down-regulated circular RNA found in ovarian carcinomas. Zhang Lei and colleagues reported that circPLEKHM3 has an important role in neoplastic initiation and progression by direct interaction with the miR9/BRCA1/DNAJB6/KLF4/AKT1 axis. Patients with loss of circPLEKHM3 expression incline to have a poor survival prognosis. Taking into account the aforementioned clinical aspects, circPLEKHM3 may be used as a novel promising prognostic and diagnostic biomarker in ovarian cancer [148]. 

Li et al. observed that the expression level of circRNA_100395 was significantly decreased in ovarian cancer tissues and cell lines when compared with noncancerous tissues and normal ovarian epithelial cell lines. Their study also indicated that lower expression of circ_100395 was highly associated with the advanced FIGO stage and diminished survival time. Moreover, the expression level of circRNA_100395 was negatively associated with the expression level of miR-1228. The upregulation of circRNA_100395 could inhibit migration, proliferation, and epithelial-mesenchymal transition (EMT) signaling pathway in ovarian cancer via modulating the miR-1228/p53/EMT axis. Consequently, circRNA_100395 might be regarded as a promising molecular biomarker in ovarian cancer therapy [149].

Functional studies performed in the last years showed that the expression levels of circ-ITCH, circPLEKHM3, and circRNA_100395 were significantly lower in ovarian cancer and validated the ceRNA hypothesis [147,149,150].

#### 5.2.4. Transfer RNA-Derived Small Non-Coding RNAs (tsncRNA)

TsncRNAs are a new category of ncRNAs with possible attributions as biomarkers in ovarian malignancies detection. TsncRNAs include tRNA halves (tiRNAs) and tRNA-derived fragments (tRFs). A study from 2019 observed using Real-Time PCR a higher expression for tRF-03357 and tRF-03358 in the serum collected from women with high-grade serous ovarian cancer and also, in ovarian cancer cell lines. Notably, tRF-03357 modulated the expression of HMBOX1, a tumor-suppressive factor, and promoted aberrant cell proliferation, migration, and invasion [148,151]. Zhou and his team found an association between a tRNA fragment termed tRF5-Glu and BCAR3 (breast cancer antiestrogen resistance 3) resulting in suppression of ovarian cancer cell multiplication [151,152].

Although no clinical diagnostic methods have been implemented using ncRNAs, numerous research studies certify their importance in the processes of detecting and monitoring ovarian cancer. Because the clinical relevance is not fully understood, clinical trials based on a large number of patients are required.

## 6. Targeted Therapy in the Personalized Management of Ovarian Cancer

In targeted therapies, patients are treated by agents targeting the alterations in tumor cells that help them grow, divide, and spread. Nowadays, in gynecological cancers, potential therapeutic targets include tumor-intrinsic signaling pathways, homologous recombination deficiency, angiogenesis, immunologic factors, and hormone receptors.

### 6.1. Inhibitors of Angiogenesis

Angiogenesis refers to the process of the formation of new vessels, and it constitutes a hallmark process of cancer progression and metastasis. The angiogenetic process is very complex and involves a large number of cytokines and associated receptors. Angiogenesis has been demonstrated to be a necessary process for oncogenesis, as well as tumor growth and dissemination through metastases. Angiogenesis is regulated by various proangiogenic and antiangiogenic factors [153].

Vascular endothelial growth factor (VEGF), a major driver of angiogenesis in many solid tumors, binds to the VEGF receptors (VEGFR, including VEGFR-1/2/3) on target cells and starts the signaling pathway using intracellular tyrosine kinases [154].

VEGF has immunosuppressive as well as proangiogenic functions, yet the impact of VEGF on local immunity and the specific mechanisms of its role in immune suppression in the tumor microenvironment remains unclear [155].

Neovasculature is considered an essential pathway for tumor growth and progression [156]. In the last years, efforts have been made to develop vascular-targeted therapies for cancer personalized treatment. Depending on this type of mechanism, vascular-targeted therapies include antiangiogenic agents and vascular-disrupting agents. Anti-angiogenesis with agents such as bevacizumab are shown to act through blocking VEGF-A action on endothelial cells. Bevacizumab belongs to a class of drugs called angiogenesis inhibitors. This drug attaches to VEGF and slows or stops cancer growth. The unseen antitumoral effects observed after bevacizumab treatment in refractory- and platinum-resistant ovarian cancer patients indicate that these responses possibly are not only caused by inhibition of angiogenesis [157].

### 6.2. PARP Inhibitors

A class of drugs called PARP inhibitors, which block the repair of DNA damage, have been found to arrest the growth of cancer cells that have pathogenic BRCA1 or BRCA2 variants.

Poly (ADP-ribose) polymerases (PARPs) are a family of related enzymes that share the function to catalyze the transfer of ADP-ribose to target proteins. PARPs play an important role in many cellular processes such as modulation of chromatin structure, transcription, replication, recombination, and DNA repair mechanisms. PARP also autoactivates itself in the presence of DNA strand breaks [158]. Because of its role in DNA repair, PARP inhibition results in genomic instability and accumulation of damaged cells in cell cycle arrest). In 2018, a landmark clinical trial showed that maintenance therapy with the PARP inhibitor olaparib benefitted women with ovarian cancer that had a pathogenic BRCA mutation [159].

The inhibition of PARP activity using dominant-negative mutant PARPs has also been shown to result in an increase in apoptosis, which arises in part due to a reduced DNA repair capacity [160]. It has been suggested that PARP is a key component of the cell cycle G2 checkpoint, which prevents a damaged cell with DNA strand breaks from being able to enter mitosis [161].

PARP inhibitors are used to treat patients with advanced ovarian cancer that has relapsed after earlier treatment. Results from three new clinical trials reveal that these drugs might also benefit women who are newly diagnosed with advanced ovarian cancer [162].

### 6.3. NTRK Inhibitors

The NTRK gene family each encodes a separate TRK protein (TRKA, TRKB, or TRKC). Functioning by the transmission of extracellular signals to the nucleus, activating survival pathways (such as the MAPK/ERK and PI3K/AKT pathways), cell growth, and proliferation [95,163]. Unless inhibited, NTRK gene fusion results in the overexpression of the TRK fusion protein. NTRK genes then tend to fuse with unrelated genes.

A very small number of ovarian cancers have changes in one of the NTRK genes. Cells with these gene changes can lead to abnormal cell growth and cancer. Larotrectinib and entrectinib are targeted drugs that stop the proteins made by the abnormal NTRK genes. These drugs can be used in patients with advanced ovarian cancer whose tumor has an NTRK gene change and is still growing despite other treatments [164].

## 7. Conclusions

Ovarian cancer is usually diagnosed in the final stages, partially due to the absence of an effective screening strategy, although, over time, numerous biomarkers have been studied and used to assess the status, progression, and efficacy of the drug therapy in this type of disorder.

Aberrant expression levels of ncRNAs were associated with different types of cancer, including ovarian malignancy. Among ncRNAs classes that were correlated with tumor initiation and progression are microRNAs, long noncoding RNAs, circular RNAs, and transfer RNA-derived small non-coding RNAs.

Even though many genetic types of research have been performed till now and several transcriptomic signatures have been proposed in ovarian cancer, there is still no clear set of specific genes involved in the process of ovarian carcinogenesis which can be used as a reference standard in carcinomas detection.

## Figures and Tables

**Table 1 biomedicines-09-00693-t001:** Type of tumors according to the WHO guidelines.

Type of Tumors According to the WHO Guidelines	Ovarian Cancer Subtypes:	Observation:	References
Serous tumors	Benign serous tumors: serous cystadenoma, adenofibroma, surface papilloma	Ovarian serous carcinomas are divided into low-grade and high-grade carcinomas, two different tumor types that have different morphology, pathogenesis, molecular events, and prognosis. Ovarian HGSCs originate from a precursor lesion on the distal fimbrial end of the fallopian tube, are associated with TP53 mutation and homologous recombination deficiency (including BRCA); LGSCs arise from the ovary from benign and borderline serous tumors, associated with BRAF and KRAS mutations.	[14,15,16]
	Borderline serous tumors: Serous borderline tumor, micropapillary variant		
	Malignant serous tumors: low-grade serous carcinoma, high-grade serous carcinoma		
Mucinous tumors	Bening mucinous tumors: Mucinous cystadenoma and adenofibroma	Such types of tumors are benign with gastrointestinal or Mullerian-type mucinous epithelium, the association of some subtypes of these tumors with dermoid cysts suggests a germ cell origin. Molecular aberration refers to copy-number loss of CDKN2A in the majority of cases, and KRAS, TP53, ERBB2, (HER2) mutations.	[14,15,16,17,18]
	Borderline mucinous tumors: Mucinous borderline tumor		
	Malignant mucinous tumors: Mucinous carcinoma		
Endometrioid tumors	Bening endometrioid tumors: Endometrioid cystadenoma and adenofibroma	The presumed tissue of origin is the endometrial epithelium, where histotype-specific mutations are present. Between these, *POLE* exonuclease domain mutations, mismatch repair deficiency, *TP53*, and non-specific molecular profile (NSMP) has been reported.	[14,15,16,19]
	Borderline endometrioid tumors: Endometrioid borderline tumor		
	Endometrioid adenocarcinoma		
	Seromucinous carcinoma		
Clear cell tumors	Benign clear cell tumors: Clear cell cystadenoma and adenofibroma	The majority of these tumors arise from transformed ovarian endometrioid lesions or benign and borderline tumors. Common mutations in ARID1A, PIK3CA, KRAS, TP53 mutations, and uncommon mismatch repair deficiency.	[14,15,16]
	Borderline clear cell tumors: Clear cell borderline tumor		
	Malignant clear cell tumors: Clear cell carcinoma		
Seromucinous tumors	Benign seromucinous tumors: Seromucinous cystadenoma and adenofibroma	According to WHO, seromucinous carcinoma is considered a subtype of endometrioid carcinoma,	[14,16,18]
	Borderline seromucinous tumors: Seromucinous borderline tumor		
	Malignant seromucinous tumors: Seromucinous carcinoma		
Brenner tumors	Benign Brenner tumors: Brenner tumor	Cell of origin of Brenner tumors is controversial; they may arise from Walthard rests which are nests of metaplasic transitional epithelium in paratubal tissue. Rare extraovarian Brenner tumors are reported, associated with a teratoma that may originate from germ cells.	[14,16,20]
	Borderline Brenner tumors: Borderline Brenner tumor		
	Malignant Brenner tumors: Malignant Brenner tumor		
Other carcinomas	Mesonephric-like adenocarcinoma	Some of these tumors arise from mesonephric remnants in the paraovarian area, or from Mullerian carcinomas that exhibit secondary mesonephric transdifferentiation. The association of mesonephric-like carcinomas with endometriosis, cystadenomas, adenofibromas, borderline tumors, and low-grade serous carcinomas was also reported. The most common molecular alterations are KRAS, NRAS, PIK3CA mutations.	[14,16,21,22]
	Undifferentiated and dedifferentiated carcinomas		
	Carcinosarcoma		
	Mixed carcinoma		
Mesenchymal tumors	Endometrioid stromal sarcoma	Some of these tumors may occur in association with another ovarian tumor, the mechanism of occurrence of these tumors may be associated with cascading-metastatic invasion of epithelial carcinomas, which spread through the bloodstream or lymphatic system, arrest at distant organ sites, and undergo extravasation into the parenchymal organ, and subsequent proliferation to form micro- and macro-metastases.	[14,22,23]
	Smooth muscle tumors		
	Ovarian myxoma		
	Other ovarian mesenchymal tumors		
Mixed epithelial and mesenchymal tumors	Mixed malignant epithelial and mesenchymal tumors, Adenocarcinoma	This type of carcinoma is composed of two or more different histological types that have a common clonal origin, which could develop through transdifferentiation of one type to another or through divergence of two histological types from a common precursor. Mixed ovarian carcinomas are rare, less than 1%, their etiology is related to the histological types, and usually are associated with endometrioid and clear cell histotypes.	[14,21,22]
Sex cord-stromal tumors (SCSTs)	Pure stromal tumors	(SCSTs) comprise a heterogeneous group of neoplasms, some of them may mimic non-SCSTs, They affect all age groups from childhood to old age and include malignancies of germ cell origin, sex cord-stromal cell origin, and a variety of extremely rare ovarian cancers. For diagnostic are used a panel of immunohistochemical markers with specificity for sex cord-stromal differentiation such as α-inhibin, calretinin, SF-1, and FOXL2, could confirm the cellular lineage of these tumors but cannot distinguish between the different histotypes within this category. Some of the molecular events linked with this tumors types are specific for some histotype, for example: in sex cord tumor with annular tubules (associated with Peutz–Jeghers sindromesyndrome) was found germline STK11 gene mutations on chromosome 19p13.3; in Steroli-Leydig cell tumor in which, patients have different hormonal manifestations with a retiform pattern or germline DICER1 mutation that occur at a younger age. SCSTs present usually wildtype for DICER1 and FOXL2 mutations. The expression of WT1, FOXL2, CD56, melan A, CD10, and CD99 also characterize many SCSTs.	[14,24,25]
	Ovarian fibroma		
	Thecoma		
	Luteinized thecoma associated with sclerosing peritonitis		
	Sclerosing stromal tumor		
	Microcystic stromal tumor		
	Signet-ring stromal tumor		
	Leydig cell tumor		
	Steroid cell tumor		
	Ovarian fibrosarcoma		
	Pure sex cord tumors		
	Adult granulosa cell tumor		
	Juvenile granulosa cell tumor		
	Sertoli cell tumor		
	Sex cord tumor with annular tubules		
	Mixed sex cord-stromal tumors		
	Sertoli-Leiding cell tumor		
	Sex cord-stromal tumor NOS		
	Gynandroblastoma		
Germ cell tumors (GCTs)	Mature teratoma	(GCTs) originate from stem cells of the early embryo and the germline. These types of tumors are characterized by the latent potency state of their cells of origin, which are reprogrammed to omnipotent, totipotent, or pluripotent stem cells. Each histotype is defined by distinct epidemiological and (epi)genomic features. These groups of tumors are rarely caused by somatic driver mutations, and molecular changes are characterized by failure to control the latent developmental potential of their cells of origin, resulting in their reprogramming. It was found that they are high sensitivity for DNA damage and are characterized by wild-type TP53 mutation.	[14,26,27]
	Immature teratoma		
	Dysgerminoma		
	Yolk sac tumor		
	Embryonal carcinoma		
	Non-gestational choriocarcinoma		
	Mixed germ cell tumor		
	Monodermal teratomas and somatic-type tumors arising from a dermoid cyst, Struma ovarii		
	Ovarian carcinoid		
	Neuroectodermal-type tumors		
	Monodermal cystic teratoma		
	Somatic neoplasms arising from teratomas: Germ cell-sex cord-stromal tumor, Gonadoblastoma		
	Mixed germ cell-sex cord-stromal tumor, unclassified		
Miscellaneous tumors	Rete cystadenoma, adenoma, and adenocarcinoma	These tumor-like ovarian lesions are histobiologically diverse, that present a wide spectrum of uncommon, varied clinical manifestations and characteristic histomorphology.	[14,28,29]
	Wolffian tumor		
	Solid pseudopapillary tumor		
	Small cell carcinoma of the ovary, hypercalcemic type		
	Wilms tumor		
Mesothelial tumors		It was demonstrated that mesothelial cells that cover the peritoneal cavity in the tumor microenvironment, cooperate with ovarian cancer cells to adhere to the peritoneum, invade, and disseminate.	[14,30]
Tumor-like lesions	Follicle cyst		[14]
	Corpus luteum cyst		
	Large solitary luteinized follicle cyst		
	Hiperreactio luteinalis		
	Pregnancy luteoma		
	Stromal hyperplasia and hyperthecosis		
	Fibromatosis and massive edema		
	Leydig cell hyperplasia		
Metastases		Ovarian metastases are malignant tumors metastasizing to the ovary from extraovarian primary site, and the pathogenesis and specific molecular events depend on the primary tumor.	[14]

**Table 2 biomedicines-09-00693-t002:** Genes expression profiling of ovarian carcinomas.

Genes Associated with Tumor Behavior	Main Role of the Gene in Carcinomatosis	References
BRCA1 and BRCA2	tumor suppressor genes, well known to play roles in hereditary breast and ovarian cancer, both BRCA1 and BRCA2 encode proteins that are involved in the repair of double-stranded DNA breaks (DSBs) by homologous recombination (HR)	[12,31,39,40,41,42,43,44]
CDKN1A	Cyclin-dependent kinase inhibitor 1A (p21, Cip1), interacting protein, encodes a protein that functions as a potent cyclin-dependent kinase inhibitor, and suffer different alteration such as missense mutations, nonsense mutations, silent mutations, and frameshift deletions and insertions.	[12,38,39,40]
HNRPA1 hnRNPs genes family	Heterogeneous nuclear ribonucleoprotein A1 are RNA-binding proteins associated with complex and diverse biological processes such as processing of heterogeneous nuclear RNAs (hnRNAs) into mature mRNAs, RNA splicing, transactivation of gene expression, and modulation of protein translation.	[39,45,46,47,48]
TP53	tumor protein p53 that acts as a tumor suppressor and regulates cell division, but these functions are context-dependent and may be influenced by numerous factors, such as cell type, microenvironment, and oncogenic events acquired during the course of tumor evolution. p53 is one of the most extensively studied proteins in cancer research.	[12,31,43,49,50,51,52,53]
DIRAS	DIRAS family, GTP-binding RAS-like, This gene encodes a member of the ras superfamily, The encoded protein acts as a tumor suppressor whose function is abrogated in many ovarian and breast cancers; DIRAS3, shares 50–60% homology to the oncogene H/N/K-RAS (DIRAS family, GTP-binding Ras-like 3) is related to ovarian and breast cancer progression.	[39,54,55,56]
BRAF, KRAS NRAS	KRAS and BRAF are involved in RAS-RAF-mitogen/extracellular signal-regulated kinase (MEK), extracellular signal-regulated kinase (ERK), and mitogen-activated protein kinase (MAPK) pathways that regulate cell proliferation. KRAS oncogene mutations exist in several histologic types of invasive epithelial ovarian carcinoma, especially stage I tumors, but are common only in tumors of mucinous histology. Mutations in BRAF and KRAS genes are the most frequent genetic aberrations found in low-grade serous ovarian carcinomas, serous borderline tumors, and mucinous cancers.	[31,43,57,58,59]
WNT2	Wingless Type MMTV integration site (WNT) gene family. Dysregulation in the WNT signaling pathway promotes or inhibits cancer biological progression.	[39,60,61,62]
IGKC	Immunoglobulin kappa constant immunoglobulin genes and proteins have been found in a variety of cancer cells, and published data suggest that Ig secreted by epithelial cancer cells can promote the growth and survival of tumor cells.	[39,53,63]
NFXL1 OZFP	NF-X1-type zinc finger protein NFXL1 named also Ovarian zinc finger protein (hOZFP), ZFHX4, ZIC2, ZNF222, ZNF143, ZNF281, FLJ13842- protein dysregulated in OVCs. Differential expressions of genes encoding the zinc finger homeobox 4 (ZFHX4) protein have been observed in different stages of OVCs. They act as a molecular regulator factor of tumor-initiating stem cells and have also DNA-binding transcription factor activity. Interacting selectively and non-covalently with zinc (Zn) ions.	[39,63,64,65]
GPCRs (G-protein-coupled receptors)	represent the largest gene family in the human genome, involved in the progression and metastasis of ovarian neoplasms, but the most important function accomplished by GPCRs, are to be drug targets, due their activities are regulated by approximately 25% of all drugs approved by the Food and Drug Administration used in OVCs therapies.	[39,66,67,68,69,70]
ferritin light chain (FTL)	encodes the light subunit of the ferritin protein, a gene that has multiple pseudogenes, involved in the rates of iron uptake and release in different tissues.	[39,71]
Other differentially expressed genes (DEGs)- associated with OVCs	S100 calcium-binding protein A1, A2 (S100A1, S100A 2), Spondin 1, (f-spondin) extracellular matrix protein (SPON1, SPOCK2), claudin (CLDN), Osteomodulin (OMD)—Bone morphogenetic protein 7 (osteogenic protein 1), Solute carrier family (SLC28A2), Spermatogenesis associated 2-like (MGC26885, Collagen, type IX, alpha 2 (COL9A2)), Solute carrier family (SLC), Brain-specific protein (CGI-38), Ki-67, cyclin B1-CDK1 complex—Cyclin-dependent kinase inhibitor, Aldehyde dehydrogenase 3 families, Ceruloplasmin (ferroxidase) CP, Homeobox D1 (HOXD1), Kallikrein family (KLK5, KLK6, KLK7, KLK8), Mesothelin (MSLN), Paired box gene 8 (PAX8), SRY (sex-determining region Y)-box, etc.	[39,42,70,72,73,74,75]

These genes can participate in OVCs development, and it was reported to have been implied in stage progression of ovarian cancer and is considered that can be used as biomarkers for prognosis. Many studies have been published regarding large-scale gene expression analyses that are differentially expressed in ovarian carcinomas, and the main goal of all of them was to identify novel diagnostic, and prognostic biomarkers in ovarian carcinoma, as well as to improved therapy and treatment of these malignancies.

## Data Availability

Not applicable.

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
