# Peer review of "Ovarian Cancer: Biomarkers and Targeted Therapy"

_biomedicines, 2021, doi:10.3390/biomedicines9060693_

Round 1

Reviewer 1 Report

nil

Reviewer 2 Report

Uterine cancer is a very subtle cancer with strong genomic instability, and it is not yet well characterized. Unfortunately, the strong phenotypic heterogeneity in the disease's progression makes staging difficult as well as the characterization of the molecular mechanisms involved. Drugs and clinical treatments reflect all this. Even other cancers behave the same way; an example is hepatocellular carcinoma. The authors also reported the most recent literature.

The authors individually report and discuss many of the genes known to be involved in this cancer, also extending to the RNAs involved, in particular miRNAs, lncRNAs, and circRNAs. They also discuss what are the more traditional markers and the newer targeted therapies.

The conclusions reflect the state-of-the-art knowledge we have of this cancer. In fact, the authors state: "there is still no clear set of specific genes involved in the process of ovarian carcinogenesis which can a reference standard in carcinomas detection."

It is the opinion of this referee that the lack of genomic or "omics" data that can place the metabolic events in space and time makes it difficult, if not impossible, to understand what are the real molecular mechanisms that involve the progression of many cancers. What is still missing in cancer studies is defining the where, the when, and the how.

The review is clearly divided into its key parts, which are well described and illustrated by adequate and recent references.

Author Response

We added more information. Please see the revised manuscript.

Round 2

Reviewer 1 Report

The author respond to review's comment and suggestion. But the language quality need improvement. Please have professional language editing service to improve it.